# Long-Term Tetrabromobisphenol A Exposure Induces Gut Microbiota Imbalance and Metabolic Disorders via the Peroxisome Proliferator-Activated Receptor Signaling Pathway in the Regenerated Gut of *Apostichopus japonicus*

**DOI:** 10.3390/biology12111365

**Published:** 2023-10-25

**Authors:** Xiaojun Song, Ying Lin, Yinfeng Zhang, Zi Wang, Xiaohan Li, Jixiang Liu, Wenwen Jiang, Jianing Chen, Linxuan Wu, Junjie Rong, Kefeng Xu, Guodong Wang

**Affiliations:** 1School of Marine Science and Engineering, Qingdao Agricultural University, Qingdao 266109, China; 2College of Medicine, Qingdao University, Qingdao 266021, China; 3Marine Science Research Institute of Shandong Province, National Oceanographic Center, Qingdao 266104, China; 4College of Food Science and Engineering, Ocean University of China, Qingdao 266003, China

**Keywords:** *Apostichopus japonicus*, TBBPA, intestine microbiota, lipid metabolism, PPAR signaling pathway

## Abstract

**Simple Summary:**

Tetrabromobisphenol A, which has been found in water, sediment, soil, household dust, human tissues, and even human milk, possesses apparent negative impacts on development and growth, increases oxidative stress, and disrupts the endocrine system. In the present study, we found that TBBPA affected the gut microbiota and intestinal health in the regenerated intestine of *Apostichopus japonicus*. TBBPA exposure reduced the enzymatic activities of superoxide dismutase, malondialdehyde and the total of antioxidant capacity. The alpha diversity indices and the relative abundance of gut microbiota decreased after TBBPA exposure. We also found that TBBPA exposure affected lipid metabolism via the PPAR signaling pathway during the process of intestinal regeneration in *A. japonicus* via transcriptome sequencing, suggesting that TBBPA exposure can affect the composition and function of gut microbiota and intestinal health in the regenerated intestine of *A. japonicus*.

**Abstract:**

Tetrabromobisphenol A (TBBPA), a commonly utilized brominated flame retardant, is found in many types of abiotic and biotic matrices. TBBPA can increase oxidative stress, disrupt the endocrine system, cause neurodevelopmental disorders and activate peroxisome proliferator-activated receptors to modulate lipid deposits in aquatic animals. However, the toxic mechanism of TBBPA on the gut microbiota and intestinal health remains unclear. *Apostichopus japonicus* is an ideal model for studying the relationship between environmental contaminants and intestinal health due to its unique capacity for evisceration and quickly regenerated intestine. In the present study, we investigated the toxic mechanism of TBBPA on the gut microbiota and intestinal health in the regenerated intestine of *A. japonicus*. The results show that TBBPA exposure decreased the health of the regenerated intestine and the enzymatic activities, alpha diversity indices, and the relative abundance of the gut microbiota. Transcriptome analysis shows that TBBPA exposure affected lipid metabolism via the PPAR signaling pathway during the process of intestinal regeneration in *A. japonicus*, suggesting that TBBPA exposure can affect the composition and function of the gut microbiota and intestinal health in the regenerated intestine of *A. japonicus*. These results provide a basis for further research on the potential toxicity of TBBPA to the intestinal health in animals.

## 1. Introduction

Tetrabromobisphenol A (TBBPA), a reliable and efficient brominated flame retardant (BFR), is widely used in furniture, plastics, textiles, and electronics [1,2]. Surprisingly, TBBPA has been found in many types of abiotic and biotic matrices, including water, sediment, soil, household dust, human tissues, and even human milk [3,4,5,6,7,8]. Therefore, the potential risks of TBBPA to human and animal health have raised concerns among toxicologists.

Intensive investigations have revealed that TBBPA possesses apparent negative properties on development and growth, increases oxidative stress, disrupts the endocrine system, induces neurodevelopmental and cardiac toxicity, and damages reproductive organs [9,10,11,12,13]. In humans, TBBPA enhances intracellular reactive oxygen species (ROS) and malondialdehyde (MDA) levels, induces mitochondrial apoptosis, and activates the NRF2 pathway in hepatocytes [14]. TBBPA also induces apoptosis by increasing ROS generation and decreasing mitochondrial membrane potential (MMP) in spermatogenic cells [11]. In mice, TBBPA affects brain neuronal development, hippocampal neurogenesis/memory retention, neurobehavior, and sensory innervation [15,16,17,18]. In zebrafish, TBBPA can affect hatchability, survival, malformation rate, and growth; TBBPA competes strongly with thyroxine (T4), inhibits the binding of triiodothyronine (T3) to thyroid hormone receptors (TRs), and affects several TR gene expressions, and its binding to peroxisome proliferator-activated receptors (PPARs) has been confirmed in vitro [19]. In addition, TBBPA could induce reproductive endocrine-disruption in the mussel (*Mytilus galloprovincialis*); induce oxidative stress, detoxification, and innate immunity in the scallop (*Chlamys farreri*); and inhibit the growth of juvenile manila clam (*Ruditapes philippinarum*) [20,21,22,23].

The gut microbiota is involved in all aspects of host organism health [24,25]. It can affect digestion and metabolism, intestinal permeability, and even immune responses when the composition and function of the gut microbiota change [26]. Recent studies have shown that the microbiome–host relationship can be modulated by chemical exposures [27]. Therefore, the microbiome has emerged as a central theme in environmental toxicology. In mice, the composition of fecal microbiome decreases during early-life exposure to TBBPA, suggesting that TBBPA can affect the composition and function of the gut microbiome [28]. Interestingly, the PPAR signaling pathway not only responds to TBBPA stress but also plays an essential role in gut microbiota homeostasis [19,29,30,31]. PPARγ activated by the gut microbiota is a homeostatic route that blocks pathogenic heterogeneous growth of *Escherichia coli* and Salmonella by decreasing the respiratory electron receptor bioavailability to Enterobacteriaceae in the colon lumen [32]. Therefore, whether the TBBPA exposure would affect the composition and function of the gut microbiome and regulate the gene expression of the PPAR signaling pathway in intestinal tissue should be studied.

The sea cucumber (*Apostichopus japonicus*), belonging to the class Holothuroidea in the phylum Echinodermata, is a commercially important aquaculture marine animal in China, Japan, and Korea, because of its valuable nutrition [33]. As a suspension and deposit feeder, sea cucumber is an ideal bioindicator to reflect the contamination of an environment [34,35]. Moreover, due to its unique capacity for evisceration and quickly gut regeneration [36], sea cucumber has gradually become an ideal model to study the relationship between intestinal health and environmental contaminants. A recent study found that TBBPA was widely distributed on the seawater surface of the Bohai Sea and Yellow Sea in China, the main areas of *A. japonicus* aquaculture, with concentrations ranging from ND (not detected) to 0.46 μg/L (about 0.85 nmol/L) in 2015 [37]. In the present study, we examined the potential toxicity of TBBPA to the regenerated intestine of *A. japonicus* to provide a basis for further study of on the relationship between TBBPA exposure and intestinal health in animals.

## 2. Materials and Methods

### 2.1. Statement of Ethics

This experiment and the use of animals were approved by the School of Marine Science and Engineering, Qingdao Agricultural University, in accordance with general national standards.

### 2.2. Experimental Animals

*A. japonicus* (50.0 ± 5.0 g) with healthy physiological condition was obtained from an aquaculture farm in Qingdao, China. The sea cucumbers were incubated for one week in our laboratory’s recirculation system at 15–19 °C, pH 7.8–8.2, and salinity of 30–32‰ to adapt them to the laboratory environment. In the experiment, we stimulated evisceration by injecting 2 mL of KCl (0.35 M) into the abdominal space of the sea cucumbers, which were randomly distributed into 9 tanks (50 × 30 × 60 cm) at a density of 7 sea cucumbers per tank.

### 2.3. TBBPA Exposure Experiment

For the TBBPA exposure experiment, the eviscerated sea cucumbers were cultured in TBBPA at concentrations of 0, 1, and 10 nmol/L. TBBPA was added to DMSO for lysis and shaken continuously. Each treatment had 3 replicates. If necessary, ultrasonic equipment was used to aid in lysis. During the experiment, we replaced half of the sea water of each tank and added TBBPA to keep the concentration constant. During TBBPA exposure, the sea cucumbers were fasted. After 28-day exposure, the coelomic fluid and regenerated intestine of 7 sea cucumbers per tank were dissected and instantly deposited in liquid nitrogen. Among them, 3 sea cucumbers were used for enzyme analysis and 16S rRNA gene sequencing, and 3 sea cucumbers were used for RNA transcriptome sequencing. Mortality was recorded daily. Because the regenerated intestine was not observed at the 10 nmol/L TBBPA exposure group after 28-day, the analysis of enzymatic activities of coelomic fluid and gut microbiota and transcriptome sequencing of regeneration intestine was only performed for 0 and 1 nmol/L TBBPA exposure groups in the subsequent analysis.

### 2.4. Enzymatic Activity Analysis

The frozen samples were thawed on ice and centrifuged at 4 °C 3000× *g* for 20 min. The supernatant was obtained and used to examine enzymatic activities within 24 h. Enzymatic activities of alkaline phosphatase (AKP, U/gprot), acid phosphatase (ACP, U/gprot), malondialdehyde (MDA, nmol/mL), superoxide dismutase (SOD, U/mL), lysozyme (LZM, U/mL), and total antioxidant capacity (T-AOC, mM) were examined using commercially available quantitative assay kits (Jiancheng Biotechnology Co., Nanjing, China) according to the manufacturer’s protocols.

### 2.5. Genomic DNA Preparation and Sequencing of 16S rRNA Gene

Genomic DNA was prepared from the regenerated intestinal contents using a commercially kit (Mo Bio Laboratories, Inc., Carlsbad, CA, USA) and detected via an agarose gel (1%) electrophoresis. The V4–V5 region in the bacterial 16S rRNA gene was used for amplification and sequencing using commercially available barcoded fusion primers. PCR was performed via 98 °C denaturation for 30 s, followed by 25 cycles (98 °C for 10 s, 53 °C for 30 s, and 72 °C for 30 s). The last extension was carried out at 72 °C for 7 min. The equimolar concentrations of the pooled amplicons were calculated and then sequencing was performed using the Illumina MiSeq platform (Illumina, San Diego, CA, USA).

### 2.6. Analysis of Intestinal Microbiota Diversity

For merging paired-end sequence reads, the FLASH 1.2.11 software (Johns Hopkins University, Baltimore, MD, USA) was used. The Fastp 0.19.5 software (https://github.com/OpenGene/fastp, accessed on 11 May 2022) was used for quality control. After quality filtering, the sequences of high-quality clean reads were analyzed using the Uparse 11 software (Independent Investigator, Tiburon, CA, USA). Sequences with a similarity sharing of 97% or greater were assigned to the same operational classification unit (OTU). OTU classification information was annotated using the RDP Classifier 2.13 algorithm (https://anaconda.org/bioconda/rdp_classifier, accessed on 1 June 2022). Multiple sequence alignment (MSA) was performed using the MAFFT 7.2 software (University of Tokyo, Chiba, Japan). Differences in phylogenetic relationships between dominant species and OTUs were determined using the MEGA X software (Tokyo Metropolitan University, Hachioji, Japan).

The alpha diversity indices of Chao1, Shannon, Simpson, and ACE, which reflect the complexity of species diversity, were analyzed using the Mothur 1.30.2 software (https://mothur.org/wiki/download_mothur/, accessed on 1 June 2022). Taxonomic abundance was analyzed using the QIIME 1.9.1 software (Northern Arizona University, Flagstaf, AZ, USA) and visualized based on the R 3.5.1 language (R Core Team, Vienna, Austria). The QIIME 1.9.1 software was also used to calculate variations in species diversities based on principal coordinate analysis (PCoA) and non-metric multidimensional scaling (NMDS). Maximum coefficients of variation were indicated by the first PCoA and second maximum coefficients of variation were indicated by the second PCoA.

### 2.7. RNA Preparation, Library Formulation, and DGE Sequencing

Total RNA was purified using TRIzol reagent (Thermo Fisher Scientific, Franklin, MA, USA) following the company’s protocol. The RNA quantity and quality of all samples were detected using 1% agarose gel and Agilent Bioanalyzer 2100 system (Agilent Technologies, Santa Clara, CA, USA). Libraries were constructed for sequencing operated using the Illumina TruseqTM RNA Sample Prep Kit for Illumina (Illumina, San Diego, CA, USA) according to the company’s protocol. The constructed libraries were used for sequencing via the Illumina HiSeq 2000 platform (Illumina, San Diego, CA, USA) and 150 bp paired-end reads were performed. Reads with low quality, adapters, and poly-N in the raw data were removed to obtain clean reads.

The reference unigenes of all samples were assembled from the *A. japonicus* reference genome (https://www.ncbi.nlm.nih.gov/genome/12044, accessed on 6 November 2017). The BLAST+ 2.9.0 software (ftp://ftp.ncbi.nlm.nih.gov/blast/executables/blast+/2.9.0/, accessed on 15 May 2022) was utilized to identify the homologous genes of reference unigenes via alignment to Swiss-Prot, the nonredundant protein (NR) and Clusters of Orthologous Groups (COG) protein databases, and the Kyoto Encyclopedia of Genes and Genomes (KEGG), Swiss-Prot. The number of reads per exon model kilobase per million mapped reads (RPKM) for each unigene was evaluated to assess the gene expression levels in the total sample. The DEGSeq R package was utilized to assess the differentially expressed genes (DEGs) in the total sample. Samples were selected if the corrected *p*-value was *p* < 0.05 and the log_2_|fold change| was more than 1.5.

### 2.8. GO and KEGG Pathway Enrichment Analyses

Gene ontology (GO) enrichment analysis of DEGs was perfected using the GOATOOLS 0.6.5 software (Fujian Agriculture and Forestry University, Fuzhou, China). Corrected *p*-value (*p* < 0.05) was considered as indicating a meaningfully enriched gene set. To further identify DEG functions and signaling pathways, KEGG database-mediated analysis for pathway enrichment based on the KEGG database was used via the KOBAS 2.1.1 software (http://kobas.cbi.pku.edu.cn/download.php, accessed on 15 May 2022). Corrected *p*-values (*p* < 0.05) for each pathway was calculated and *p* < 0.05 was judged as indicating meaningfully enriched pathways in the DEGs.

### 2.9. RNA Preparation and Real-Time Quantitative Polymerase Chain Reaction (qRT‒PCR)

Total RNA was prepared from the regenerated gut and a reverse transcriptase was used to generate cDNA using the PrimeScript™ RT Reagent Kit (Vazyme Bio Inc., Nanjing, China). The cycles were carried out as 40 cycles of 30 s at 95 °C, 15 s at 60 °C, and 10 min at 72 °C. The primers for quantification of the β-actin gene and the PPAR signaling pathway genes in this experiment are shown in Appendix A. Data were quantified based on the Ct values utilizing the 2^−ΔΔCt^ protocol. Each value was considered significant at *p* < 0.05.

### 2.10. Statistical Analysis

One-way analysis of variance (ANOVA) or *t*-test were applied to analyze differences in survival, enzymatic activities in the antioxidant- and immune-systems, and intestinal bacterial diversity in the regenerated gut of sea cucumbers among the TBBPA treatments. The assumption of homogeneity of variance was checked before conducting ANOVA, and the LSD method was used for post hoc test if the ANOVA result was significant. All data were expressed as mean ± SD unless otherwise specified. The SPSS 23.0 software (SPSS Inc., Chicago, IL, USA) was used for all statistical analyses.

## 3. Results

### 3.1. Analysis of Survival Rate and Regenerated Intestine Morphology

The survival rate of *A. japonicus* exposed to TBBPA concentrations of 0, 1, and 10 nmol/L for 28 days was 100.0, 100.0, and 95.2%, respectively. The survival rate was reduced, but there was no significant difference (One-way ANOVA, F_[2]_ = 1.00, *p* = 0.422, Appendix A). Interestingly, there were entire regenerated intestines in the 0 nmol/L and 1 nmol/L TBBPA exposure groups, but none in the 10 nmol/L TBBPA-exposure group (Figure 1).

### 3.2. Enzymatic Activities in Antioxidant and Immune Systems in Regenerated Intestine

The results show that the enzymatic activities of malondialdehyde (MDA) (*t*_[2]_ = 7.114, *p* = 0.019), superoxide dismutase (SOD) (*t*_[2]_ = 8.647,*p* = 0.013), and total antioxidant capacity (T-AOC) (*t*_[2]_ = 7.101,*p* = 0.019) significantly decreased after TBBPA challenge (Figure 2A–C). The activity of lysozyme (LZM) (*t*_[2]_ = 4.513,*p* = 0.049) significantly increased, while the activities of alkaline phosphatase (AKP) (*t*_[2]_ = 0.198,*p* = 0.86) and acid phosphatase (ACP) (*t*_[2]_ = 0.341,*p* = 0.766) did not significantly change after the TBBPA exposure (Figure 2D–F).

### 3.3. Microbial Diversity Decreased in the Regenerated Intestines of A. japonicus

The results regarding microbial diversity in the regenerated intestines of *A. japonicus* show that the Chao1 diversity index (*t*_[2]_ = 35.793,*p* < 0.001), the Shannon diversity index (*t*_[2]_ = 15.103,*p* = 0.004), and the ACE diversity index (*t*_[2]_ = 32.258,*p* < 0.001) decreased while the Simpson diversity index (*t*_[2]_ = 4.706,*p* = 0.049) increased after TBBPA stimulation, indicating that the microbial diversity of the regenerated intestines of *A. japonicus* decreased after TBBPA stimulation (Figure 3). The microbial composition data demonstrate that the dominant species in the regenerated intestine of *A. japonicus* were Proteobacteria (82.42%), Planctomycetota (6.27%), Verrucomicrobiota (3.87%), Armatimonadota (2.79%) and Bacteroidota (1.95%) in the 0 nmol/L TBBPA exposure group and Proteobacteria (77.26%), Firmicutes (29.97%), Armatimonadota (0.42%) and Bacteroidota (0.17%) in the 1 nmol/L TBBPA-exposure group (Figure 4A). After TBBPA exposure, the relative amount of Firmicutes (*t*_[2]_ = 37.264,*p* < 0.001) increased, and the relative amount of Planctomycetota (*t*_[2]_ = 45.283,*p* < 0.001), Verrucomicrobiota (*t*_[2]_ = 30.959, *p* = 0.0013), Armatimonadota (*t*_[2]_ = 37.264,*p* < 0.001) and Bacteroidota (*t*_[2]_ = 25.136,*p* = 0.002) decreased (Figure 4B–F).

The differences in microbial composition in the regenerated intestine of *A. japonicus* after the TBBPA challenge were evaluated via β-diversity analysis. The PCoA analysis demonstrated that the microbial composition in the regenerated intestine of *A. japonicus* was different after TBBPA challenge, with the first principal locus accounting for 54.82% and the second principal locus accounting for 37.89% of the total variance (Appendix A). The results of the NMDS analysis were similar to those of the PCoA analysis (Appendix A).

### 3.4. The Gene Expression Related to Metabolism Decreased in the Regenerated Intestine of A. japonicus

Profiles of RNA sequencing of high-throughput gene expression showed that the pairwise sequence alignment of all samples was >97%, suggesting that the RNA-seq sequences were highly homologous to the reference genome sequence of *A. japonicus* (Appendix A). In the control and 1 nmol/L TBBPA treatment group, 19,413 and 19,742 gene models were found, respectively. Of these, 17,977 genes were expressed in both groups. An analysis of differentially expressed genes (DEGs) revealed that a total of 369 genes were differentially expressed after the TBBPA challenge, in which the expression of 118 genes and 251 genes increased and decreased, respectively (Figure 5A and Appendix A). PCA analysis was then performed to detect differences between the TBPA-treated and the control groups. The results showed that there was a difference in expression between the control group and the 1 nmol/L TBPA-treated group, with the first and second principal loci accounting for 35.20% and 22.22%, respectively (Figure 5B). To identify the functional mechanism of DEGs, an analysis of Gene Ontology (GO) annotation was performed. Three broad categories, including molecular functions, cellular components, and biological processes, were identified. It was found that TBBPA could affect enzymatic activities and metabolic levels in the regenerated intestine of *A. japonicus* after TBBPA exposure (Figure 5C).

Genes often exert their functions through gene networks and signaling pathways. Therefore, we performed a KEGG enrichment analysis to explore the molecular mechanisms by which TBBPA affects *A. japonicus* intestinal regeneration. The results showed that a total of 202 DEGs mapped onto 33 pathways in the KEGG database (*p* < 0.05), and the top 20 of the 33 KEGG pathways are shown in Figure 6A. Further analysis of these pathways led to their consolidation into five categories: metabolism, organismal systems, environmental information processing, human diseases, and cellular processes (Figure 6B). Lipid metabolism dominated the metabolic categories, including arachidonic acid, linoleic acid, glycerophospholipid, and α-linolenic acid metabolisms. (Figure 6A,C). In organismal systems, the digestive system was predominant, including digestion and absorption of vitamins, carbohydrates, and fats (Figure 6A,D). In environmental information processing, signal transduction systems, including the PPAR pathway and the AMPK pathway, were predominant (Figure 6A,E). These results suggested that TBBPA might affect lipid metabolism, biological processes, and signal transduction during gut regeneration in *A. japonicus*. An analysis of DEGs in metabolism, organismal systems, and environmental information processing revealed that several genes might be involved in different biological processes (Figure 6F). The heatmap results showed that most DEGs were downregulated, including nsLTP (nonspecific lipid transporter), BHMT1 (betaine-homocysteine S-methyltransferase), and ENPP7 (ectonucleotide pyrophosphatase/phosphodiesterase 7). The expression of GSULT (galactosylceramide sulfotransferase) and P450 increased after the TBBPA challenge. Protein interactions in the DEG network analysis showed that TBBPA affected the Wnt signaling pathway and lipid metabolism during gut regeneration of *A. japonicus* (Figure 7). In conclusion, the microbial diversity and gene expression for metabolism were decreased in the regenerated gut of *A. japonicus* after long-term TBBPA stress.

### 3.5. PPAR Signaling Pathway Genes Play Vital Roles in the Regenerated Intestine of A. japonicus

The results showed that nine genes of the PPAR signaling pathway were downregulated and one gene of the PPAR signaling pathway was upregulated (Figure 8 and Appendix A). The spearman’s rank correlation analysis and Pearson correlation coefficient indicated that the PPAR signaling pathway gene expressions were significantly correlated between the DEGs and qRT-PCR data (Spearman’s rank correlation: ρ = 0.673, *p* = 0.033; Pearson correlation coefficient: r = 0.899, *p* < 0.001). These outcomes indicate that PPAR signaling pathway genes play essential roles in the regenerated intestine of *A. japonicus* after TBBPA stimulation.

## 4. Discussion

The regenerative capacity of tissue is typical in echinoderms [38]. *A. japonicus* is able to regenerate its gut after being eviscerated due to attacks from natural enemies or sudden alterations in the circumstances [39,40]. Recently, with the increase in TBBPA in the marine environment, the effects on intestinal health in animals are still unknown. Herein, we explored the composition and distribution of the intestinal microbiota in *A. japonicus* exposed to TBBPA, as well as gene expression during the intestinal regeneration. The results reveal that TBBPA induces intestinal microbiota imbalance and metabolic abnormalities via the PPAR signaling pathway during the intestinal regeneration of *A. japonicus*.

At low concentrations, TBBPA has been shown to be toxic to growth, development, and oxidative stress in aquatic animals [9,13,23,41,42]. Recent works have demonstrated that TBBPA has a dose-dependent activity on hatchability, survival, malformation rate, growth rate, and oxidative stress in zebrafish [9]. In addition, TBBPA can cause craniofacial and oocyst deformities, and shortening of the tail and curvature of the spine in zebrafish [13]. The present study showed that TBBPA inhibited the regeneration of intestine in *A. japonicus*. There were no significant differences in survival after the TBBPA challenge, but two *A. japonicus* died in the 10 nmol/L groups, and none died in the 0 and 1 nmol/L groups. This might be because, during the gut regeneration process, the TBBPA exposure period was not long enough to affect the survival of *A. japonicus*. Therefore, further study is needed to determine the effect of TBBPA on the viability of *A. japonicus*. MDA is an indicator of endogenous oxidative damage and an end product of lipid peroxidation. The results show significant differences in T-AOC enzymatic activity and SOD and MDA levels in the TBBPA-challenge group, suggesting that TBBPA may induce oxidative stress in the regenerated intestine of *A. japonicus*.

The gut microbiota refers to microorganisms that populate the gastrointestinal tract and exhibit symbiosis with their host [43,44]. The gut microbiota has been identified as being involved in host metabolism, immune defense, development, and evolution, and propensity to a variety of noninfectious and infectious diseases [45]. A recent study indicated that the regenerated intestinal tissues of *A. japonicus* had a different microbial community composition with higher alpha diversity [46]. The restoration of the bacterial community consists of two stages with different assembly mechanisms, starting with a bacterial consortium in the remnant gastrointestinal tract and tending to establish a unique microbiota in the later stages of restoration [46,47]. In addition, the regenerated intestinal tissues of *A. japonicus* contain a microbiome that appears to provide energetical and immunological benefits to the host reaction, including carbon fixation and microorganisms that degrade invading pathogens [46]. However, exposure to TBBPA can affect the richness and diversity of the community in the human gut [48]. In the present study, we found that TBBPA exposure reduced the alpha diversity and microbial composition in the regenerated intestine of *A. japonicus*.

Although cellular events and signaling pathways associated with intestinal regeneration in *A. japonicus* have been reported [40,49], there is a lack of knowledge regarding the association between intestinal regeneration and TBBPA in *A. japonicus*. Previous works have demonstrated that the Wnt/β-catenin signaling pathway is activated during early gut regeneration in *A. japonicus*, suggesting that the Wnt pathway is activated during the gut regeneration process in *A. japonicus* [50,51,52,53]. Furthermore, the cellular dedifferentiation observed during intestinal regeneration can be separated from cell proliferation events, and these cellular processes can be modulated by inhibitors and activators of specific signaling pathways [40]. In the current work, we revealed that the expression of Wnt pathway genes decreased after the TBBPA challenge, suggesting that TBBPA may downregulate the gene expressions related to the Wnt pathway during intestinal regeneration in *A. japonicus*. Recent works have demonstrated that the PPAR signaling pathway can mediate TBBPA-induced biological processes such as the intracellular affinity effects of nuclear receptors and apoptosis [54,55]. The PPAR signaling pathway can also regulate the transcriptional regulation of lipid metabolism [56,57]. In addition, the AMPK signaling pathway can regulate the expression of PPARα-driven genes via binding to regulatory regions of genes for lipid and glucose metabolism, which are co-regulated by PPARα and GRα [58,59]. In the present study, the KEGG enrichment analysis revealed that TBBPA affected the metabolic pathways of lipids, including ether lipid, linoleic acid, and glycerophospholipid metabolisms. Interestingly, DEGs induced by TBBPA during the progression of *A. japonicus* gut regeneration were enriched in the PPAR signaling pathway and the AMPK signaling pathway. These results suggest that TBBPA induces metabolic abnormalities via the PPAR signaling pathway in the regenerated intestine of *A. japonicus*. These results provide a theoretical basis for additional extensive studies on the effects of TBBPA toxicity on intestinal heath in animals.

## 5. Conclusions

In summary, we found that TBBPA affected the gut microbiota and intestinal health in the regenerated intestine of *A. japonicus*. TBBPA exposure reduced the enzymatic activities of superoxide dismutase, malondialdehyde, and the total of antioxidant capacity. The alpha diversity indices and the relative abundance of gut microbiota decreased after TBBPA exposure. TBBPA exposure affected lipid metabolism via the PPAR signaling pathway during the process of intestinal regeneration in *A. japonicus*, as revealed by transcriptome sequencing. These results suggested that TBBPA exposure can affect the composition and function of gut microbiota and intestinal health in the regenerated intestine of *A. japonicus*.

## Figures and Tables

**Figure 1 biology-12-01365-f001:**
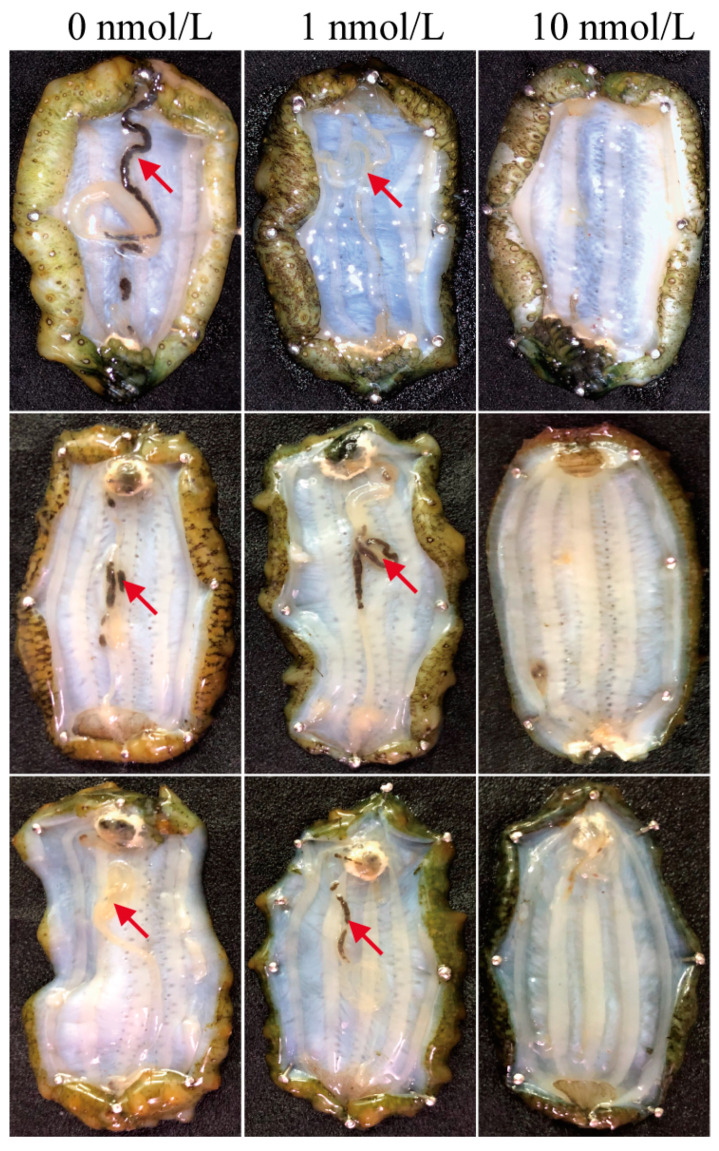
Morphology of the regenerated intestines after TBBPA exposure for 28 days. There are 3 samples that are presented for each concentration. Red arrows indicate the regenerated intestine.

**Figure 2 biology-12-01365-f002:**
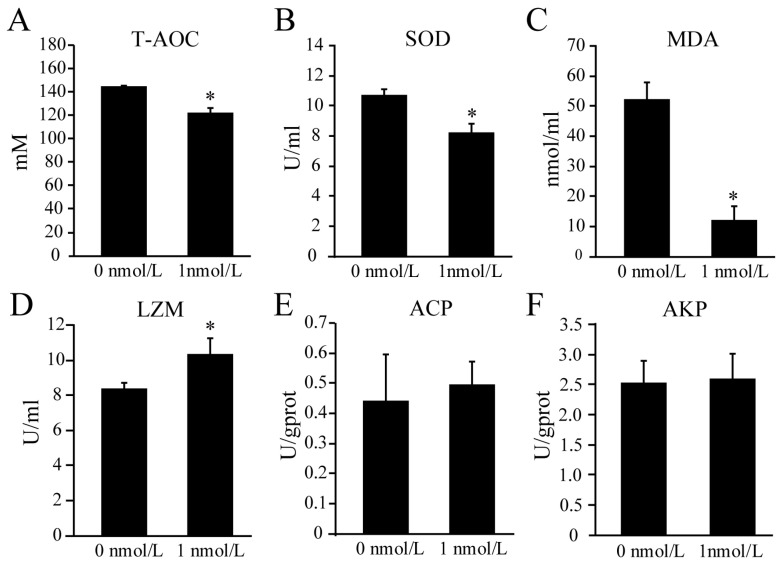
The activity of antioxidant- and immune-related enzymes in coelomic fluid of *A. japonicus* after TBBPA exposure. (**A**) The activity of T-AOC of *A. japonicus* after TBBPA exposure. (**B**) The activity of SOD of *A. japonicus* after TBBPA exposure. (**C**) The activity of MDA of *A. japonicus* after TBBPA exposure. (**D**) The activity of LZM of *A. japonicus* after TBBPA exposure. (**E**) The activity of ACP of *A. japonicus* after TBBPA exposure. (**F**) The activity of AKP of *A. japonicus* after TBBPA exposure. * presents *p* < 0.05 (*t*-test).

**Figure 3 biology-12-01365-f003:**
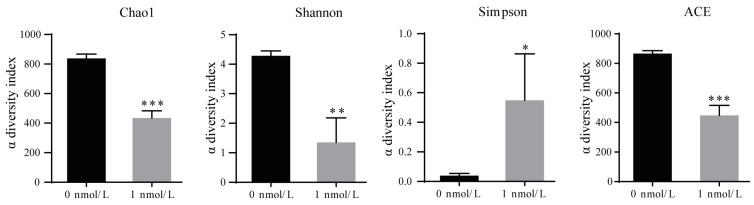
Alpha diversity indexes in the regenerated intestine of *A. japonicus* after TBBPA exposure. * presents *p* < 0.05; ** presents *p* < 0.01; *** presents *p* < 0.001 (*t*-test).

**Figure 4 biology-12-01365-f004:**
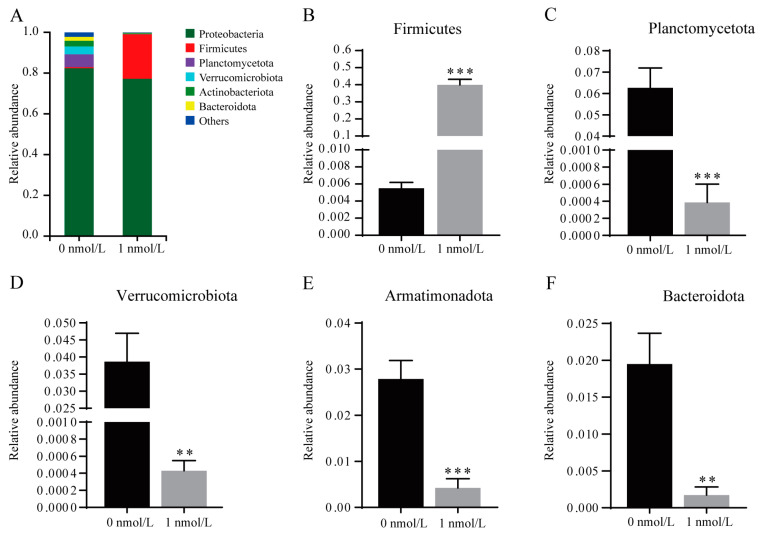
Microbial diversity and components in the regenerated intestine of *A. japonicus*. (**A**) Relative abundance (at the phylum level) of the gut microbiota in the regenerated intestine of *A. japonicus* after TBBPA treatment. (**B**–**F**) Relative abundance of Firmicutes (**B**), Planctomycetota (**C**), Verrucomicrobiota (**D**), Armatimonadota (**E**) and Bacteroidota (**F**) after TBBPA treatment. ** presents *p* < 0.01; *** presents *p* < 0.001 (*t*-test).

**Figure 5 biology-12-01365-f005:**
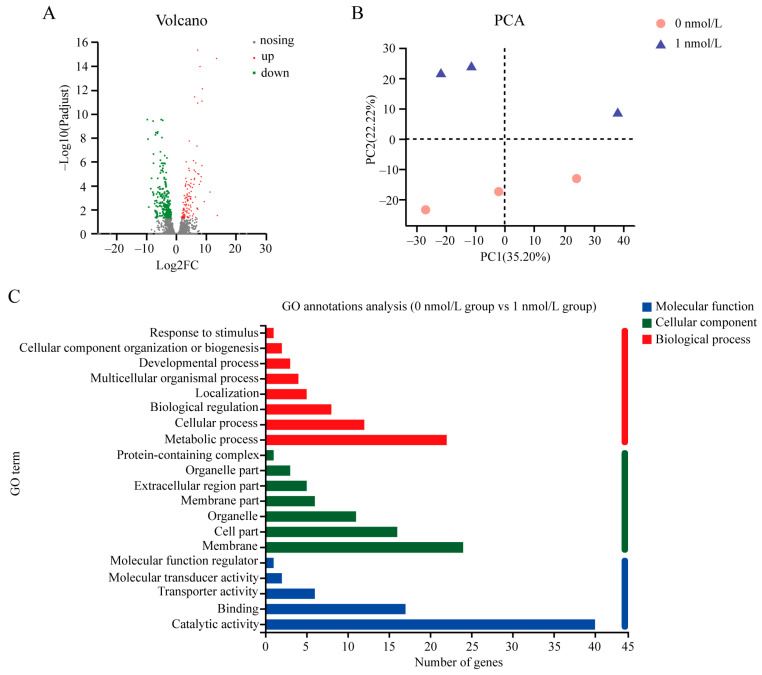
The differentially expressed genes (DEGs) in the regenerated intestine of *A. japonicus* after TBBPA treatment. (**A**) Volcano plotting of the number of DEGs in the regenerated intestine of *A. japonicus* following TBBPA treatment. (**B**) Differential gene expression in the regenerated intestine of *A. japonicus* treated with different concentrations of TBBPA. (**C**) GO annotation analysis of DEGs in the regenerated intestine of *A. japonicus* after TBBPA challenge.

**Figure 6 biology-12-01365-f006:**
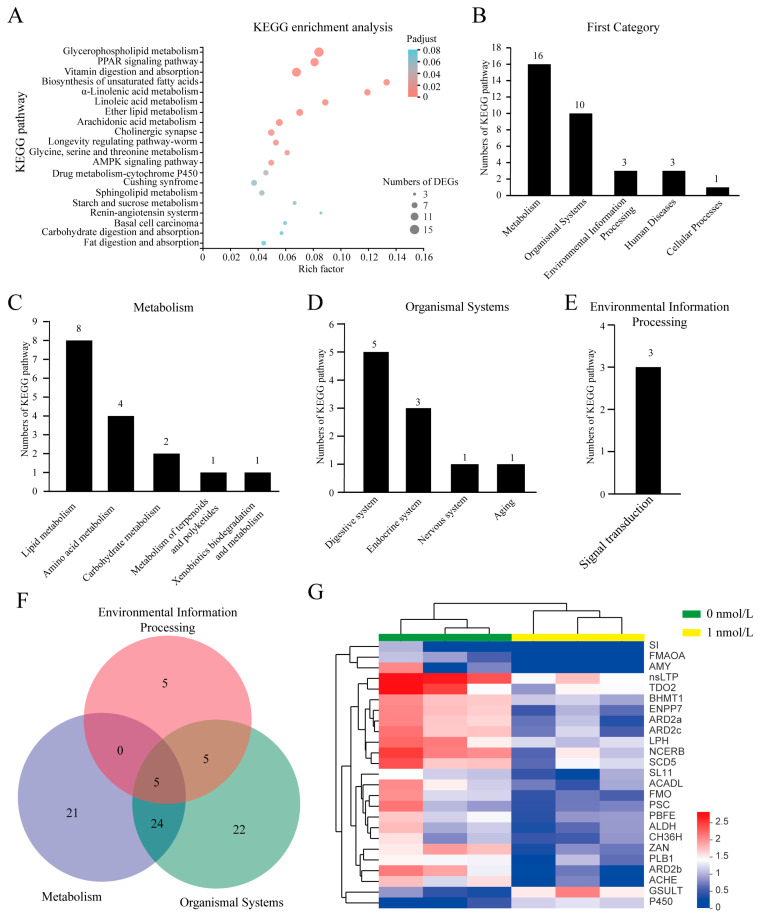
KEGG pathway enrichment analysis of differentially expressed genes (DEGs) in the regenerated intestine of *A. japonicus* after TBBPA exposure. (**A**) KEGG pathways involved in DEGs and the top 20 pathways out of 33 KEGG pathways. (**B**) The number of KEGG pathways in the first category. The number of KEGG pathways for metabolism (**C**), organismal systems (**D**), and environmental information processing (**E**) in the second category. The number on the column of (**B**–**E**) represents the number of KEGG pathways. (**F**) The number of DEGs involved in metabolism, biological systems, and environmental information processing shown as a Venn diagram. (**G**) Heat map analysis of DEGs involved in metabolic processes.

**Figure 7 biology-12-01365-f007:**
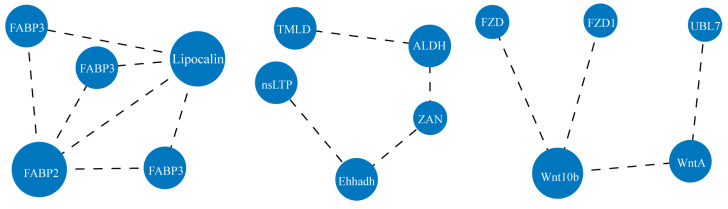
Co-expression network of DEGs in the regenerated intestine of *A. japonicus* after TBBPA exposure.

**Figure 8 biology-12-01365-f008:**
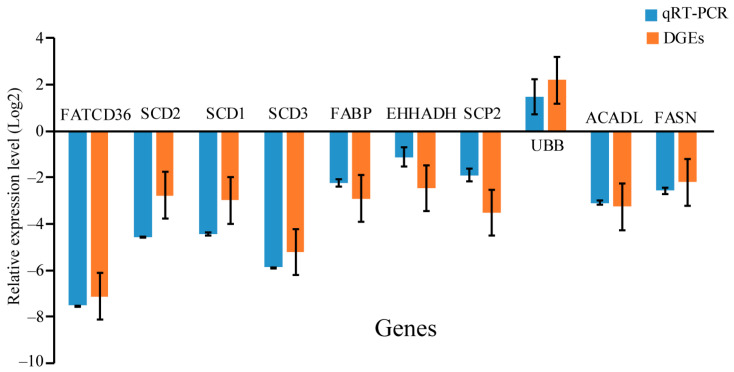
Expression levels of PPAR signaling pathway genes based on qRT-PCR and DGE data, where the Y- and X-axes indicate gene expression fold change and gene names, respectively.

## Data Availability

The raw data supporting the conclusions of this article will be made available by the authors, without undue reservation.

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
