# Peer review of "Long-Term Tetrabromobisphenol A Exposure Induces Gut Microbiota Imbalance and Metabolic Disorders via the Peroxisome Proliferator-Activated Receptor Signaling Pathway in the Regenerated Gut of Apostichopus japonicus"

_biology, 2023, doi:10.3390/biology12111365_

Round 1
Reviewer 1 Report
Xiaojan Song and fellow researchers conducted a study revealing that Tetrabromobisphenol A (TBBPA) has the capability to influence both the gut microbiota and the overall intestinal well-being in regenerated intestines of A. japonicus. This influence was observed through a decrease in the activity of enzymes associated with total antioxidant function, specifically superoxide dismutase and malondialdehyde. Additionally, exposure to TBBPA resulted in alterations to diversity indices and the relative abundance of gut microbiota, particularly impacting processes related to lipid metabolism.
The topic is the demand for today’s time and will be interested to the readers of the journal. It is well-written, mostly easy to follow, and figures are well put together. However, I have following concerns.
1. What concentration TBBPA can influence the human gut microbiota?
2. Can one use knock-down or knock out system to further illustrate the signaling-associated with gut microbiota in A. japonicus?
Reviewer 2 Report
As a typical emerging contaminant, TBBPA has been widely detected in coastal areas. However, the toxic effects and mechanism of TBBPA on marine invertebrate are less clarified. This study investigated the effects of environmentally related concentration of TBBPA on the intestinal health of the sea cucumber, and they found that TBBPA induced gut microbiota imbalance and metabolic disorders via PPAR signaling pathway in the regenerated gut. Generally, it is an interesting study, and the results can provide a basis for elucidating the toxic mechanism of TBBPA on the intestinal health in aquatic animals. However, there are some comments should be addressed, especially some of the M&M should be described in detail.
Major comments
1. Introduction section
The research progress about the toxicity of TBBPA on the aquatic animals especially marine animals should be briefly described. Please add some related research summary.
2. Materials and Methods section
The description of the exposure experiment section should be detailed introduced.
Did you renew the sea water and TBBPA during the exposure? If so, how long did you change the water and add the TBBPA? Moreover, what was the final concentration of DMSO used in the exposure groups, and did this concentration of DMSO have an effect on the sea cucumber? Finally, did you feed the sea cucumbers during the exposure? Please specify
3. 2.4 Enzyme activity analysis
The activity unit of these enzymes including AKP, ACP, MDA, SOD, LZM and T-AOC should be briefly described.
Minor comments
L18, L23, intestinal should be intestine
L18, enzymes activity should be enzymatic activities
L29,33, mechanism of TBBPA toxicology should be toxic mechanism of TBBPA
L47, delete (fresh and saltwater)
L71, works should be plays
L73, E. coli should be Escherichia coli
L78, Apostichopus japonicus should be in italic
L84, toxicology should be contaminants
L88, delete exposure, intestinal should be intestine
L101, ml should be mL
L111, g should be in italic
L115, verified should be examined
L116, corresponding to should be according to
L120, what was the name of the kit, please specify
L173, why the β-actin gene was selected as the internal control, have you ever checked the stability or do you have some references to support this selection?
L203, different should be changed
L204, function for should be impair?
L255, the sentence should be: It was found that TBBPA could affect enzymatic activity and metabolic levels in the regenerated intestine after TBBPA challenge in A. japonicus
L256-257, repetitive description, delete suggesting that .....intestinal.
L288, digestive organs? it should be biological process not organs
L292-293, the full name of BHMT1, ENPP7 and GSULT should be shown.
L332, zebrafish oxidative stress should be oxidative stress in zebrafish
L333, it is better to show in which kind of animal
L339-340, MDA ......peroxidation, rewrite this sentence
L350, two stages are better than 2 stages
Some of the sentences should be carefully checked.
